# Large-Scale and Multi-Perspective Opinion Summarization with Diverse Review Subsets

**Han Jiang, Rui Wang, Zhihua Wei,** *Yu Li, Xinpeng Wang*
Department of Computer Science and Technology, Tongji University, Shanghai, China
`{2230780, rwang, zhihua_wei, 2331891, wangxinpeng}@tongji.edu.cn`

## Abstract

Opinion summarization is expected to digest larger review sets and provide summaries from different perspectives. However, most existing solutions are deficient in epitomizing extensive reviews and offering opinion summaries from various angles due to the lack of designs for information selection. To this end, we propose SUBSUMM, a supervised summarization framework for large-scale multi-perspective opinion summarization. SUBSUMM consists of a review sampling strategy set and a two-stage training scheme. The sampling strategies take sentiment orientation and contrastive information value into consideration, with which the review subsets from different perspectives and quality levels can be selected. Subsequently, the summarizer is encouraged to learn from the sub-optimal and optimal subsets successively in order to capitalize on the massive input. Experimental results on `AmaSum` and `Rotten Tomatoes` datasets demonstrate that SUB-SUMM is adept at generating *pros*, *cons*, and *verdict* summaries from hundreds of input reviews. Furthermore, our in-depth analysis verifies that the advanced selection of review subsets and the two-stage training scheme are vital to boosting the summarization performance.

## 1 Introduction

A plethora of online resources has been appealing for techniques of automatic information mining. Opinion summarization, as a task of generalizing views from a group of documents (e.g., reviews, posts, and discussions) related to an entity and presenting them in text form, has received considerable attention. The summarization of user opinions is of great advantage to public opinion research, marketing analysis, and decision-making (Im et al., 2021). While circumventing the tedious document-by-document browsing, it also offers more significant details compared to a single sentiment rating (Wang and Wan, 2021).

Due to the burgeoning amount of online reviews and user needs, opinion summarization is expected to (1) process larger sets of documents and (2) provide summaries from different perspectives. One mainstream solution models the cross-document relations with sentence or document representations, using mean function (Chu and Liu, 2019; Bražinskas et al., 2020b; Li et al., 2020), convex combination (Iso et al., 2021), and graph (Erkan and Radev, 2004; Ganesan et al., 2010) as well as other hierarchical structures (Isonuma et al., 2021; Amplayo et al., 2021a). These approaches are proven to achieve remarkable results with a moderate amount of reviews, usually within 10 (Shapira and Levy, 2020); however, they perform unsatisfactorily when the number of reviews further increases, as they focus on the fusion rather than the selection of the information. Another solution concatenates the reviews for long-range language models (Beltagy et al., 2020; Zaheer et al., 2020; Mao et al., 2022; Pang et al., 2023) and Large Language Models (LLMs; OpenAI, 2023), which converts multi-document summarization into single-document summarization (Bražinskas et al., 2020a; Oved and Levy, 2021; Ke et al., 2022; Brazinskas et al., 2022; Bhaskar et al., 2023). Despite the benefit brought by the LLMs, these methods struggle to handle the overlong combined reviews, missing a step to select from them either. Bražinskas et al. (2021) first proposes to select smaller subsets of input reviews and provides verdict, pros, and cons summaries, yet differentiated treatments of different perspectives are not reflected in their method. Limited by data, there are seldom works targeting large-scale and multi-perspective opinion summarization.

To address the problems, we propose SUB-SUMM, a supervised summarization framework for large-scale and multi-perspective opinion summarization. SUBSUMM comprises a review sampling strategy set and a two-stage training scheme. The

---
*Corresponding author

review sampling strategies are formulated with sentiment analysis and contrastive information valuation. With different strategies, the review subsets from different angles and quality levels can be selected. Then, the two-stage training method enables the summarization model to learn from the sub-optimal and optimal review subsets successively to fully utilize the input reviews within the model capacity. During the training stage II, a contrastive loss term is incorporated to further boost the performance of the summarizer.

By coupling with SUBSUMM, the Pre-trained Language Model (PLM) outperforms previous state-of-the-art models and LLMs under zero-shot settings on the `AmaSum` and `Rotten Tomatoes` datasets in our experiments, which demonstrates the superiority of the proposal. Further analysis also proves the effectiveness of the two modules in SUBSUMM.

The contributions of this paper are as follows.

- We propose a large-scale opinion summarization framework[1] to address the challenge of summarizing large review sets and providing opinions from different perspectives by selecting valuable review subsets.

- We present (1) a review sampling strategy set based on sentiment analysis and contrastive information valuation and (2) a two-stage training scheme promoting the digestion and absorption of the massive input.

- We substantiate the effectiveness of the proposed opinion summarization framework SUBSUMM with sufficient experiments and in-depth analysis on two opinion summarization datasets from different domains.

## 2 Related Work

**Opinion Summarization** As high-quality annotation for the large opinion corpora is expensive to obtain (Ge et al., 2023), most works of opinion summarization are unsupervised, summarizing a limited number of reviews. Among the abstractive approaches, VAE-based and synthetic-dataset-based models have the upper hand.

The VAE-based models (Chu and Liu, 2019; Bražinskas et al., 2020b; Li et al., 2020; Iso et al.,

---

[1]The code is available at https://github.com/Salomeeeee/SubSumm.

2021; Isonuma et al., 2021) summarize through the aggregation of the latent representations of the reviews. COOP (Iso et al., 2021) considers the convex combination of input review representations. These methods work well with fewer reviews, while they suffer a performance drop when processing numerous reviews.

The synthetic-dataset-based methods (Amplayo and Lapata, 2020; Bražinskas et al., 2020a; Oved and Levy, 2021; Wang and Wan, 2021; Amplayo et al., 2021b; Ke et al., 2022; Brazinskas et al., 2022) transform the unsupervised task into a supervised task by constructing review-summary pairs from original data. PASS (Oved and Levy, 2021) applies systematic perturbations to the input reviews for more candidate summaries and trains a classifier to rank the candidates. CONSISTSUM (Ke et al., 2022) measures the distances between reviews from aspect, sentiment, and semantics to create highly relevant review-summary pairs. ADA-SUM (Brazinskas et al., 2022) first fine-tunes the PLM with a synthetic dataset, then performs fine-tuning in a few-shot manner. The idea of making full use of the original text is embodied thoroughly in these methods.

Benefiting from the growth of annotated data for opinion summarization, there are some emergent studies on supervised methods. Bražinskas et al. (2021) provide a large-scale opinion summarization dataset enabling supervised training. They formulate the task as jointly learning to select informative reviews and summarize the opinions, and their solution SELSUM is based on reinforcement learning (REINFORCE; Williams, 1992). Aiming at avoiding the challenges brought by reinforcement learning, we decouple the process of selection and summarization in this work.

**Contrastive Learning** Contrastive learning in automatic summarization (Cao and Wang, 2021; Xu et al., 2021; Sun and Li, 2021; Liu and Liu, 2021; Liu et al., 2022) also gives us much inspiration. CLIFF (Cao and Wang, 2021) creates negative samples with automatically generated erroneous summaries. SIMCLS (Liu and Liu, 2021) trains an extra model with contrastive learning to evaluate and rank the candidate summaries. BRIO (Liu et al., 2022) introduces contrastive learning to assign a dual role to the model, alleviating inference performance degradation. In this work, we explore contrastive learning for multi-document summarization rather than single-document sum-

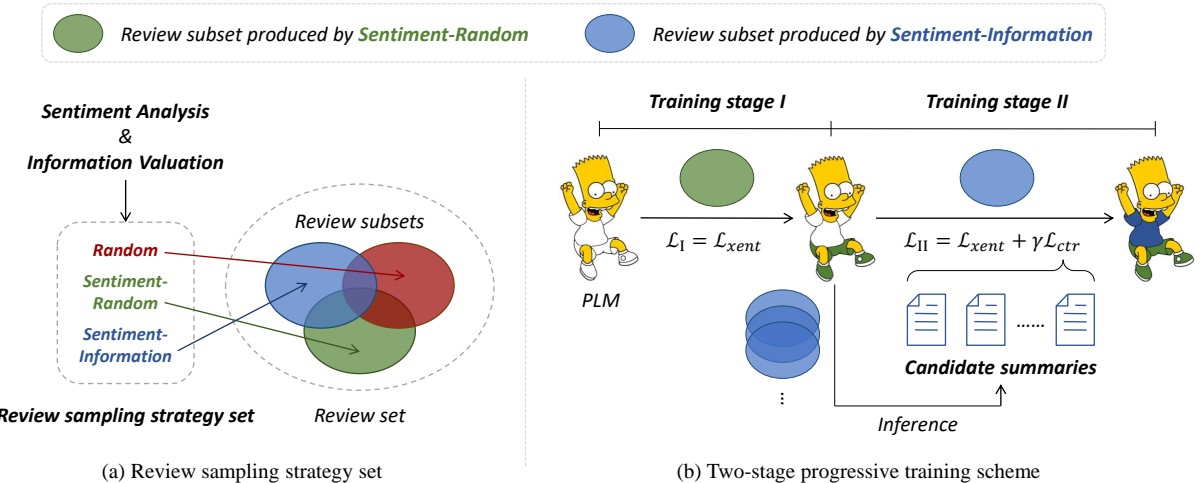

(a) Review sampling strategy set

(b) Two-stage progressive training scheme

Figure 1: The framework of SUBSUMM. (a) The review sampling strategy set based on sentiment analysis and information valuation. The three strategies, *Random Sampling*, *Sentiment-Random Sampling*, and *Sentiment-Information Ranking* are abbreviated as *Random*, *Sentiment-Random*, and *Sentiment-Infomation*. (b) The two-stage training scheme, under which the PLM (BART (Lewis et al., 2020) in this work) learns to summarize sub-optimal and optimal subsets successively.

marization, and a PLM is fine-tuned contrastively for the information valuation.

## 3  Methodology

We introduce SUBSUMM, a supervised framework for large-scale and multi-perspective opinion summarization, as illustrated in Fig. 1. SUBSUMM is composed of a review sampling strategy set regarding sentiment orientation and information value, as elucidated in Sec. 3.1; and a two-stage training scheme, where contrastive learning with candidate summaries is extra performed, see Sec. 3.2.

Given the entity set and the corresponding sample set, for every sample $\{R_{1:N}, S\}$, the goal of opinion summarization is to learn a function $f$ that takes the review set as input and outputs a summary as close as possible to the reference:

$$S \leftarrow f(R_{1:N}) \qquad (1)$$

where $R_{1:N}$ is the original review set, and $S$ is the reference opinion summary. This paper mainly discusses the situation where $R_{1:N}$ is too large to be processed with most language models. For review set $R_{1:N}$, let $R_{1:K}$ be the review subset where $K \ll N$.

### 3.1  Review Sampling Strategy Set

**Sentiment Analysis**  We leverage sentiment analysis to filter the reviews roughly. It is supposed that the reviews with similar sentiment orientations are less likely to conflict in content than those with contrary sentiment orientations. On the other hand, the sentiment tendency of the summary can be controlled by adjusting the proportion of input reviews with different sentiments; in this way, multiple angles are formed.

We formulate sentiment analysis as a text classification task. The sentiment tags of reviews in $R_{1:N}$ are computed as:

$$p_i^{sen} = SLM(R_i) \qquad (2)$$
$$Sen_i = \arg\max_j p_i^{sen}(j) \qquad (3)$$

where $SLM(\cdot)$ is a PLM with a linear classification head, and $p_i^{sen}$ refers to the probability distribution over the sentiment classes of the review $R_i$. The class with the highest probability is taken as the sentiment tag $Sen_i$. We use the rating of each review in the dataset as the sentiment label and apply a negative log likelihood loss w.r.t. the sentiment distribution $p_i^{sen}$. After fine-tuning, the sentiment tags of all the reviews can be obtained.

**Contrastive Information Valuation**  Information valuation has finer granularity than sentiment analysis. Intuitively, once a subset of reviews is selected for summarization, the closer the generation is to the reference, the more valuable the information in the subset may be.

Given the reference summary, the information value of an input review is tied to its similarity with the reference; ROUGE (Lin, 2004) is an appropriate metric to estimate such similarity. Thus we fit the ROUGE score of a review

$R_i = \{r_i^{(1)}, ..., r_i^{(|R_i|)}\}$ by modeling the correlation between the review and the whole review set:

$$h_i^{(1)}, ..., h_i^{(|R_i|)} = Enc(r_i^{(1)}, ..., r_i^{(|R_i|)}) \quad (4)$$

$$h_i = \frac{1}{|R_i|} \sum_{k=1}^{|R_i|} h_i^{(k)} \quad (5)$$

$$Corr_i = \frac{1}{N-1} \sum_{1 \le j \le N, j \ne i} h_i h_j \quad (6)$$

where $Enc(\cdot)$ is a transformer (Vaswani et al., 2017) encoder, and $h_i^{(k)}$ denotes the last hidden state of token $r_i^{(k)}$. The review representation $h_i$ is computed by averaging the last hidden states of the tokens in $R_i$. $Corr_i$ is the correlation score between $R_i$ and the review set $R_{1:N}$. We refer to the *leave-one-out* setting in unsupervised opinion summarization, computing $Corr_i$ by the dot product of $h_i$ and the mean representation of the others.

Mindful of the volume of the original review set, it is unfeasible to fit the distribution of the ROUGE score directly or employ list-wise loss. Therefore, we resort to a contrastive margin loss:

$$\mathcal{L} = \sum_{i=1}^{N} \sum_{r(j)>r(i)} \max(0, Corr_j - Corr_i + \lambda_{ij}) \quad (7)$$

where $r(i)$ accounts for the ranking of $R_i$ when sorted by ROUGE($R_i, S$) in descending order, and $\lambda_{ij} = \lambda(r(j) - r(i))$ is a margin varying with the rankings, defined following Zhong et al. (2020); Liu and Liu (2021); Liu et al. (2022). The pairwise loss allows the model to learn the ROUGE rankings of a large review set. After fine-tuning, we can get the estimated information values of all the reviews.

**Multi-level Review Sampling Strategies** Drawing support from the sentiment analysis and the contrastive information valuation, diverse review sampling strategies can be formed to select $R_{1:K}$ out of $R_{1:N}$. We find that it is not ideal to accomplish the task with a single optimal subset, which will be explained in Sec. 4.3. To tackle the problem, we introduce stochastic factors to develop sampling strategies of multiple quality levels.

The sampling strategy set consists of the following three strategies:

- **Random Sampling**: Randomly sample $K$ reviews from the original review set $R_{1:N}$ as the subset $R_{1:K}$.

- **Sentiment-Random Sampling**: Firstly, divide all reviews into positive and negative types according to their sentiment tags $Sen_i$. Secondly, the number of reviews of each type in $R_{1:K}$ is determined by the type of the reference summary:

$$(K^+, K^-) = \begin{cases} (K, 0), & pros \\ (0, K), & cons \\ (\frac{KN^+}{N}, \frac{KN^-}{N}), & verdict \end{cases} \quad (8)$$

where $(K^+, K^-)$, $(N^+, N^-)$ stands for the numbers of positive and negative reviews in $R_{1:K}, R_{1:N}$; $K^+ + K^- = K, N^+ + N^- = N$. Finally, randomly sample $K^+, K^-$ reviews from the positive and negative types respectively for $R_{1:K}$.

- **Sentiment-Information Ranking**: Firstly, compute $(K^+, K^-)$ likewise. Secondly, sort the reviews in descending order by the estimated information value $Corr_i$ in two types separately. Finally, take the top-$K^+$ positive reviews and the top-$K^-$ negative reviews for $R_{1:K}$.

The quality of the corresponding review subsets should improve in sequence.

### 3.2 Two-Stage Training for Large-Scale Opinion Summarization

SUBSUMM embodies a two-stage training scheme encouraging the summarizer to learn from the sub-optimal and optimal review subsets successively.

In stage I, we choose the sub-optimal strategy, *Sentiment-Random Sampling* to re-sample the review subset $\dot{R}_{1:K}$ at each training epoch and train the model with standard maximum likelihood estimation (MLE):

$$\theta^* = \arg\max_{\theta} \sum_i \log p_\theta(S|\dot{R}_{1:K}^{(i)}) \quad (9)$$

where $\theta$ denotes the parameters of the abstract model, and $p_\theta$ represents the probability derived by the parameters. The cross entropy loss is defined over the reference sequence of length $l$ as:

$$\mathcal{L}_I = \mathcal{L}_{xent} =$$

$$-\sum_{i=1}^{l} \sum_{s \in \mathcal{V}} p^*(s|\dot{R}_{1:K}, S_{<i}) \log p_\theta(s|\dot{R}_{1:K}, S_{<i}) \quad (10)$$

where $s$ can be any token in the vocabulary $\mathcal{V}$, and $p^*$ refers to an one-hot distribution. $S_{<i}$ stands for a pre-defined start token and the first $i-1$ tokens of the reference summary. However, Standard MLE is prone to *exposure bias* since it heavily relies on the ground-truth sequence (Zhang et al., 2019). Meanwhile, whichever strategy is adopted, the reviews sampled are only a part of the original review set, where the information can be further exploited.

In stage II, we take a cue from the practice of assigning probability mass to candidate summaries during training (Liu et al., 2022). Theoretically, assigning probability mass to a summary means an opportunity for the summary to pass on knowledge to the model through backpropagation. Hence the range of probability mass allocation is essentially the range of model learning, and better candidate summaries ought to compete for more probability mass. We plan to reuse the original review set via the candidate summaries.

To start with, we slightly modify the optimal strategy (i.e., *Sentiment-Information Ranking*), as some perturbations are required to obtain various candidate summaries for contrastive learning:

- **Sentiment-Information Ranking (modified)**: After computing $(K^+, K^-)$ in Eq. 8, take the estimated information value $Corr_i$ of each review as weight to sample $K^+, K^-$ reviews from the positive and negative types severally.

Next, the modified optimal strategy is repeatedly conducted to get $M$ review subsets, with which $M$ candidate summaries $\hat{S}_1, \hat{S}_2, ..., \hat{S}_M$ are generated by the model from stage I. The review subset produced by the original optimal strategy, denoted by $\ddot{R}_{1:K}$, will be the training input. We again calculate the ROUGE scores of the reviews in $\ddot{R}_{1:K}$ with the reference summary $S$ to derive the rankings and apply a contrastive loss term similar to Eq. 7:

$$\mathcal{L}_{ctr} = \sum_{i=1}^{M} \sum_{r(j)>r(i)} \max(0, Lh_j - Lh_i + \lambda_{ij}) \tag{11}$$

where $Lh_i$ is the length-normalized likelihood of the candidate summary $\hat{S}_i$, which is defined following Liu et al. (2022):

$$Lh(S) = \frac{\sum_{i=1}^{|S|} \log p_\theta(s_i|\ddot{R}_{1:K}, S_{<i})}{|S|^\alpha} \tag{12}$$

Here $\alpha$ is a length penalty hyperparameter. This term enforces the model to assign more probability mass to better candidate summaries.

Finally, to maintain the generation ability of the pre-trained model, we follow Edunov et al. (2018) to use the multi-task loss:

$$\mathcal{L}_{II} = \mathcal{L}_{xent} + \gamma \mathcal{L}_{ctr} \tag{13}$$

where $\gamma$ is the weight of the contrastive loss term. By involving the candidate summaries in training, stage II raises the utilization rate of the original review set and alleviates the problem of *exposure bias*; it acts as a complement to stage I considering the addition of the review subsets with higher quality and the contrastive loss term.

During inference, given a review set $R_{1:N}$, SUB-SUMM predicts the sentiment tag and information value of each review with the fine-tuned PLMs in Sec. 3.1, then selects the optimal review subset $R_{1:K}$ according to the *Sentiment-Information Ranking* strategy and summarizes the subset using the summarization model from stage II.

## 4 Experiments

### 4.1 Experimental Settings

**Datasets** We choose two opinion summarization datasets with large review sets as our testbed. The statistics are shown in Appendix A.

AmaSum[2] (Bražinskas et al., 2021) is a product review dataset where each sample contains a large number of reviews and reference summaries written by professional reviewers. Unlike other datasets, AmaSum provides reference summaries from three perspectives, namely *verdict*, which is equivalent to general opinion summary; *pros* and *cons*, which summarize the most important positive and negative details. As shown in Table 6, with 4.2k tokens on average, the combined reviews in AMASUM are too long to summarize with most summarizers. We refer to the preprocessing in SEL-SUM but split the dataset into three partitions with different targets.

Rotten Tomatoes[3] (RT; Wang and Ling, 2016) is a large-scale movie review dataset. For each movie, a one-sentence critic consensus is constructed by an editor to summarize the opinions in professional critics, which is treated as the reference summary. We follow Amplayo et al. (2021b) to preprocess the dataset; the data in RT and the *verdict* partition of AmaSum are equally treated in our experiments.

---

[2]https://github.com/abrazinskas/SelSum
[3]https://web.eecs.umich.edu/~wangluxy/data.html

Table 1:

| Method | Pros | | | Cons | | | Verdict | | |
|---|---|---|---|---|---|---|---|---|---|
| | R-1 | R-2 | R-L | R-1 | R-2 | R-L | R-1 | R-2 | R-L |
| *Unsupervised* | | | | | | | | | |
| MEANSUM[†] | 10.44 | 0.63 | 9.55 | 5.95 | 0.45 | 5.29 | 13.78 | 0.93 | 11.70 |
| LEXRANK[†] | 14.12 | 1.50 | 12.81 | 8.28 | 0.82 | 7.24 | 15.12 | 1.84 | 12.60 |
| COPYCAT[†] | 15.12 | 1.48 | 13.85 | 6.81 | 0.82 | 5.89 | 17.05 | 1.78 | 14.50 |
| EXTSUM[†] | 19.06 | 2.47 | 17.49 | 11.63 | 1.19 | 10.44 | 18.74 | 3.01 | 15.74 |
| *Supervised* | | | | | | | | | |
| SELSUM[†] | 21.29 | 4.00 | 19.39 | 14.96 | 2.60 | 13.07 | 24.33 | **5.29** | 18.84 |
| LONGFORMER | 22.40 | 4.71 | 15.36 | 14.68 | 2.53 | 11.62 | 22.56 | 4.83 | 17.08 |
| SELSUM* | 23.17 | 4.77 | 21.13 | 15.12 | 2.83 | 13.07 | 22.87 | 4.85 | 18.05 |
| BRIO | 25.48 | 4.58 | 23.50 | 16.65 | 2.94 | 14.60 | 24.93 | 4.78 | 19.44 |
| SUBSUMM | **26.25** | **4.96** | **24.18** | **16.72** | **3.00** | **14.80** | **25.36** | 5.04 | 19.58 |
| *Zero-Shot* | | | | | | | | | |
| QG | 18.40 | 2.21 | 16.02 | 13.27 | 1.39 | 11.61 | 14.75 | 1.23 | 12.06 |
| CHATGPT | 18.53 | 2.37 | 14.68 | 13.92 | 1.78 | 11.93 | 22.88 | 3.50 | **19.79** |

Table 1: Automatic evaluation results on `AmaSum` dataset. Best models are shown in bold and 2nd best models are underlined; † means that the results are copied from Bražinskas et al. (2021). We retrained SELSUM* on *pros*, *cons*, and *verdict* separately for a fair comparison.

| Method | R-1 | R-2 | R-L |
|---|---|---|---|
| *Unsupervised* | | | |
| W2VCENT[†] | 13.93 | 2.10 | 10.81 |
| LEXRANK[†] | 14.88 | 1.94 | 10.50 |
| OPINOSIS[†] | 14.98 | 3.07 | 12.19 |
| MEANSUM[†] | 15.79 | 1.94 | 12.26 |
| SNCENT[†] | 15.90 | 2.01 | 11.74 |
| BERTCENT[†] | 17.65 | 2.78 | 12.78 |
| DENOISESUM[†] | 21.26 | 4.61 | 16.27 |
| *Weakly Supervised* | | | |
| PLANSUM[†] | 21.77 | 6.18 | 16.98 |
| *Supervised* | | | |
| BRIO | 23.72 | 5.16 | 18.05 |
| LONGFORMER | **24.96** | 6.34 | 18.40 |
| SUBSUMM | **24.96** | **6.66** | **19.08** |
| *Zero-Shot* | | | |
| QG | 18.14 | 2.34 | 14.28 |
| CHATGPT | 22.73 | 4.21 | 17.52 |

Table 2: Automatic evaluation results on `RT` dataset. Best models are shown in bold and 2nd best models are underlined; † means that the results are copied from Amplayo et al. (2021b).

**Baselines** Concerning the baselines, we select a series of competitive models for the two datasets.

On `AmaSum` dataset, the baselines include (1) unsupervised extractive models LEXRANK (Erkan and Radev, 2004) and EXTSUM (Bražinskas et al., 2021); (2) unsupervised abstractive models MEAN-SUM (Chu and Liu, 2019) and COPYCAT (Bražinskas et al., 2020b); (3) supervised abstractive models SELSUM, LONGFORMER (Beltagy et al., 2020), and BRIO; and (4) zero-shot solutions related to LLMs, including GPT-3.5-turbo (CHATGPT) as well as QG (Bhaskar et al., 2023) based on QFSumm (Ahuja et al., 2022) and GPT-3 (Brown et al., 2020).

On `RT` dataset, the extra baselines are (1) unsupervised extractive models W2VCENT (Rossiello et al., 2017), SNCENT (Amplayo and Lapata, 2020), and BERTCENT (Amplayo et al., 2021b); (2) unsupervised abstractive models OPINOSIS (Ganesan et al., 2010) and DENOISESUM (Amplayo and Lapata, 2020); (3) weakly supervised model PLANSUM (Amplayo et al., 2021b). We classify PLANSUM as a weakly-supervised summarizer since it uses additional information other than review text.

A detailed introduction to the baselines is in Appendix B.

**Implementation Details** We used RoBERTa-base (Liu et al., 2019) for the sentiment analysis, a BART-base (Lewis et al., 2020) encoder for the contrastive information valuation, and BART-base

| Summary | Pros | | | Cons | | | Verdict | | |
|---|---|---|---|---|---|---|---|---|---|
| | Info↑ | Coh↑ | N-R↑ | Info↑ | Coh↑ | N-R↑ | Info↑ | Coh↑ | N-R↑ |
| GOLD | **19.7** | 5.7 | 6.4 | 0 | 0 | 4 | 6.9 | 5.2 | **10.2** |
| SELSUM | -20.2 | 2.9 | **11.8** | -6.3 | -0.6 | -0.9 | -7.2 | **6.5** | 0.6 |
| BRIO | -7.5 | -25.6 | -23.5 | -5.1 | -7.3 | -10.7 | -8.4 | -17.2 | -14.7 |
| SUBSUMM | 8 | **17** | 5.3 | **11.4** | **7.9** | **7.6** | **8.6** | 5.5 | 3.9 |

Table 3: Human evaluation results on AmaSum. Info, Coh, and N-R are abbreviations of *Informativeness*, *Coherence*, and *Non-Redundancy*.

| Method | AmaSum | | | |
|---|---|---|---|---|
| | Pros | Cons | Verdict | **RT** |
| SUBSUMM | 0.66 | 0.67 | 0.89 | 0.70 |
| CHATGPT | 0.34 | 0.33 | 0.11 | 0.30 |

Table 4: Win rates of SUBSUMM and CHATGPT in the pair-wise comparisons.

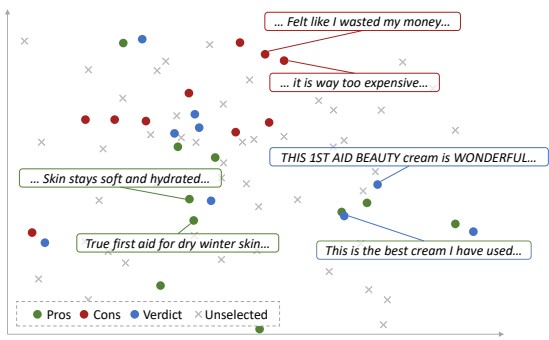

Figure 2: A t-SNE (van der Maaten and Hinton, 2008) plot of the embeddings of most reviews from a product in AmaSum, where the dots of different colors represent the reviews selected for different perspectives. A few outliers are omitted.

as the backbone of our summarizer and its variants. In all of the review sampling strategies, we selected $K = 10$ reviews for every subset, which is explained in Appendix D. All the following experiments were conducted on 2 Geforce RTX 3090 GPUs. For the hyperparameters and more details, please refer to Appendix C.

## 4.2 Results

**Automatic Evaluation** We used ROUGE-1/2/L as the evaluation metrics and reported the F-1 scores. For AmaSum, we evaluated *pros*, *cons*, and *verdict* separately. As shown in Table 1 and Table 2, SUBSUMM significantly outperforms other methods on both datasets. Specifically, there are

two observations:

(1) SUBSUMM excels in generating summary with obvious emotion tendency, i.e., *pros* and *cons*. We notice that the three targets in AmaSum are equally treated by all the baselines, indicating the lack of exploration into the difference between the perspectives. SUBSUMM samples only positive/negative reviews for pros/cons summary. As depicted in Fig. 2, the reviews sampled for *pros* and *cons* are distributed in different zones of the semantic space, with the reviews for *verdict* evenly scattered between them. It not only reduces inconsistencies, but also adds valuable information to the input, since positive reviews always point out more advantages of the product, and vice versa.

(2) The supervised methods score generally higher than the LLM-related methods, and SUBSUMM has an edge over the congener supervised systems. Although LLMs possess strong versatility in text generation, fine-tuning standard PLMs on annotated data seems non-trivial for opinion summarization. Compared with the other supervised methods, SUBSUMM reuses the reference summaries through contrastive learning in both information valuation and the training stage II, thus comprehensively utilizing the annotations. Especially, LONGFORMER's sparse attention mechanism plays an implicit selection role, while the review sampling strategies of SUBSUMM consider the sentiment tendency and information value, which is more sophisticated and task-specific.

**Human Evaluation** As a supplement to automatic evaluation, we conducted a user study using Best-Worst Scaling (BWS; Louviere et al., 2015) detailed in Appendix E. The four evaluated summaries were GOLD (reference) summary and summaries generated by SELSUM, BRIO, and SUBSUMM. The three criteria were *Informativeness*, *Coherence*, and *Non-Redundancy*.

Results in Table 3 demonstrate the considerable

| Method | Pros | | | Cons | | | Verdict | | |
|---|---|---|---|---|---|---|---|---|---|
| | R-1 | R-2 | R-L | R-1 | R-2 | R-L | R-1 | R-2 | R-L |
| *Sampling Strategy* | | | | | | | | | |
| RAND | 21.83 | 4.28 | 19.91 | 14.06 | 2.55 | 12.35 | 22.67 | 4.66 | 17.81 |
| SENTI-RAND | 22.33 | 4.41 | 20.45 | 14.92 | 2.75 | 13.13 | 23.06 | 4.79 | 17.98 |
| SENTI-INFO | 22.82 | 4.60 | 20.80 | 14.42 | 2.48 | 12.59 | **23.51** | 5.03 | 18.16 |
| SENTI-RAND-INFO | **23.01** | **4.76** | **21.02** | **15.46** | **2.92** | **13.60** | **23.51** | **5.04** | **18.32** |
| *Training Scheme* | | | | | | | | | |
| SUBSUMM | **26.25** | **4.96** | **24.18** | **16.72** | **3.00** | **14.80** | **25.36** | **5.04** | **19.58** |
| w/o Stage I | 25.48 | 4.58 | 23.50 | 16.65 | 2.94 | 14.60 | 24.93 | 4.78 | 19.44 |
| w/o Stage II | 23.01 | 4.76 | 21.02 | 15.46 | 2.92 | 13.60 | 23.51 | **5.04** | 18.32 |
| RAND in Stage I | 26.03 | 4.93 | 23.96 | 16.66 | 2.97 | 14.78 | 24.71 | 4.93 | 19.43 |
| RAND in Stage II | 25.95 | 4.95 | 24.02 | 16.50 | 2.77 | 14.54 | 24.60 | 4.96 | 19.45 |

Table 5: Results of analysis experiments on `AmaSum`. R-1/2/L are the ROUGE-1/2/L F-1 scores, and the highest scores in both blocks are shown in bold.

practical value of our model. Regarding *Informativeness*, summaries from SUBSUMM display comparable, even more information than GOLD summaries. As for *Coherence*, SUBSUMM leaves the users the best reading experience with correct grammar and straightforward expression. In terms of *Non-Redundancy*, SUBSUMM does not present the most succinct summaries, but considering the first two criteria, the redundancy is still acceptable.

We further compared our model with the LLM via 50 head-to-head tests between SUBSUMM and CHATGPT. The test cases were randomly sampled from the two datasets (15 samples from each partition in `AmaSum`'s test set and 5 samples from `RT`'s test set), and the annotators were asked to make pair-wise comparisons without the reference summaries. The results are shown in Table 4. It seems that the users prefer the summaries generated by SUBSUMM. An obvious issue of CHATGPT is that it cannot control the output length within a few calls when the input is overlong. Consequently, most of the summaries generated are either excessively long or abruptly truncated with the maximum length argument fixed. In addition, though CHATGPT is qualified to produce fluent text, it suffers from more severe hallucination than our model, which may compromise its ROUGE scores. In Table 10 are some supporting cases.

## 4.3 Analysis

With the purpose of deeper insights into our proposal, we carried out some in-depth analysis experiments on `AmaSum`, using the same metrics as in automatic evaluation. We also reported the results on `RT` in Appendix F.

**Comparison between Sampling Strategies** As aforementioned, the quality of the three strategies in the review sampling strategy set would elevate sequentially. To confirm, we compare summarizers from the training stage I with different sampling strategies in the upper block of Table 5. RAND, SENTI-RAND, and SENTI-INFO apply *Random Sampling*, *Sentiment-Random Sampling*, and *Sentiment-Information Ranking* in training and inference respectively; SENTI-RAND-INFO is trained with *Sentiment-Random Sampling* but infers on review subsets produced by *Sentiment-Information Ranking*.

By comparing RAND with SENTI-RAND, it can be seen that with the aid of sentiment analysis, the review subsets sampled appear more useful for the summaries with emotion tendencies. There is no clear improvement from SENTI-RAND to SENTI-INFO, so we add SENTI-RAND-INFO to ascertain the reason. SENTI-RAND-INFO and SENTI-RAND only differ in the test input, while the former wins with a clear margin, suggesting *Sentiment-Information Ranking* produces better review subsets. SENTI-RAND-INFO shares the same test input with SENTI-INFO but results in higher ROUGE scores, possibly because the stochastic factor prevents the potential over-fitting problem. It also drops a hint that employing diverse review subsets might promote the model performance.

**Insight into Two-Stage Training Scheme** We investigate the gains from the two-stage training scheme through an ablation study. The variants in the bottom block of Table 5 share the same test input with SUBSUMM.

Our experiments evidence that both stage I and stage II are significant to model performance, while the latter plays a greater role. We suppose that the two stages are complementary to each other: the standard MLE training in stage I functions as task-specific initialization, and the multi-task learning in stage II passes on more knowledge to the model, mitigating the *exposure bias* problem.

Moreover, we explore how the two-stage training scheme contributes to the summary quality by replacing the sub-optimal and optimal strategies in the two training stages with *Random Sampling*. It leads to observable performance decreases, yet they are slighter than those when stage I or II is directly removed. It can be inferred that besides the complementary training objectives and additional training steps, the sensible selection of the review subsets is also conducive to model training.

## 5   Conclusion

In this paper, we put forward a supervised summarization framework for large-scale and multi-perspective opinion summarization, SUBSUMM. SUBSUMM supports a two-stage training scheme based on a set of review sampling strategies of multiple quality levels. Our model surpasses the state-of-the-art models and LLM-related systems on `AmaSum` and `RT`, manifesting its superiority in dealing with plentiful reviews and displaying various points of view. The analysis experiments verify that both components of SUBSUMM help the summarizer achieve better results.

In the future, we are planning to (1) explore more review sampling strategies to fully learn the aspect information and (2) combine the proposed framework with LLMs and generalize it to other large-scale multi-input tasks.

## Limitations

There are also some limitations in SUBSUMM. In Table 1, the *verdict* partition, the ROUGE-2 F1-score of our model does not outweigh that of SEL-SUM; the ROUGE-L F1-score of our model is slightly lower than that of CHATGPT.

Firstly, since ROUGE-2 reflects 2-gram recall, we suspect that this is due to the absence of explicit designs for aspect learning in SUBSUMM, which causes the model to miss more 2-gram aspect terms than SELSUM (We noticed that SELSUM emphasizes aspect learning). Secondly, ROUGE-L is computed based on the longest common subsequence, which has something to do with the fluency of the generation. We find that there are some errors, like repetitions and incomplete first words in the summaries from SUBSUMM. Compared to the LLMs with extensive parameters, our proposal still has room for improvement in language modeling.

## Ethics Statement

We used only publicly available datasets, artifacts, and figures. Nevertheless, we realize that the proposed framework may produce fabricated and potentially harmful contents, for the PLMs used are pre-trained on heterogeneous web corpora. Therefore, we recommend that the users cautiously apply the proposal and its by-products.

## Acknowledgements

The work is partially supported by the National Nature Science Foundation of China (No. 61976160, 61906137, 61976158, 62076184, 62076182) and Shanghai Science and Technology Plan Project (No. 21DZ1204800) and Technology Research Plan Project of Ministry of Public and Security (Grant No. 2020JSYJD01).

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

| AmaSum | Train | Dev | Test |
|---|---|---|---|
| Entity | 26,660 | 3,302 | 3,362 |
| Reviews per Ent. | 77.78 | 77.76 | 76.80 |
| Tokens per Rev. | 53.98 | 54.13 | 53.97 |
| Tokens per Sum. | | | |
| - Pros | 39.13 | 39.21 | 38.81 |
| - Cons | 18.34 | 18.31 | 17.81 |
| - Verdict | 21.72 | 21.97 | 21.59 |
| **Rotten Tomatoes** | Train | Dev | Test |
| Entity | 2,453 | 536 | 737 |
| Reviews per Ent. | 74.39 | 74.85 | 73.26 |
| Tokens per Rev. | 26.52 | 26.32 | 26.41 |
| Tokens per Sum. | 23.80 | 23.59 | 23.82 |

Table 6: Statistics of the preprocessed datasets. The average numbers are taken except for "Entity".

## A   Dataset Statistics

The statistics of the datasets after preprocessing are shown in Table 6. The average numbers are taken except for "Entity".

## B   Baselines

On `AmaSum` dataset, the baselines are unsupervised extractive models (a) LEXRANK (Erkan and Radev, 2004), a PageRank-like algorithm that extracts sentences based on graph centrality, (b) EXTSUM (Bražinskas et al., 2021), which uses the same ROUGE greedy heuristic as in Liu and Lapata (2019); unsupervised abstractive models (c) MEANSUM (Chu and Liu, 2019), which generates opinion summary by reconstructing the mean of review embeddings, (d) COPYCAT (Bražinskas et al., 2020b), a VAE summarizer with hierarchical continuous latent representations to model products and individual reviews; supervised abstractive models (e) SELSUM, a model jointly learns to select informative subsets of reviews and summarizing the opinions, (f) LONGFORMER, a long-range model with an attention mechanism that scales linearly with sequence length, (g) BRIO, the state-of-the-art model of general abstractive summarization; and LLM-related solutions (h) GPT-3.5-turbo, (i) QG (Bhaskar et al., 2023), a pipeline where reviews are summarized by QFSumm (Ahuja et al., 2022) and GPT-3 (Brown et al., 2020), specifically the text-curie-001 model successively.

On `RT` dataset are some other baselines: unsupervised extractive models (j) W2VCENT (Rossiello et al., 2017), (k) SNCENT (Amplayo and Lapata,

| | AmaSum | | | |
|---|---|---|---|---|
| Hyperparameter | Pros | Cons | Verdict | **RT** |
| $bsz$ I | 16 | 16 | 16 | 8 |
| $bsz$ II | 16 | 16 | 16 | 16 |
| $warmup$ I | 5,000 | 5,000 | 5,000 | 500 |
| $warmup$ II | 3,000 | 3,000 | 3,000 | 300 |
| $\gamma$ (Eq. 13) | 0.1 | 1.0 | 0.1 | 0.1 |
| $lenpen$ | 0.5 | 0.5 | 1.0 | 1.0 |
| $minlen$ | 35 | 25 | 25 | 30 |

Table 7: Hyperparameter setting. $bsz$ I, II denote the batch sizes in the two training stages. $minlen$ stands for the minimum generation length in the beam search algorithm.

2020), and (l) BERTCENT (Amplayo et al., 2021b), which take encodings from word2vec (Mikolov et al., 2013), LSTM-based model (Radford et al., 2017), and BERT (Devlin et al., 2019) as the input representations; unsupervised abstractive models (m) OPINOSIS (Ganesan et al., 2010), a graph-based model that leverages token-level redundancy to summarize text, (n) DENOISESUM (Amplayo and Lapata, 2020), which re-formulates the summarization task as a denoising task; and a weakly supervised model (o) PLANSUM (Amplayo et al., 2021b), which constructs the synthetic dataset with a Dirichlet distribution parametrized by a content planner.

## C   Implementation Details

The codes we used for fine-tuning the PLMs in sentiment analysis, contrastive information valuation, and the training stage I were implemented with Fairseq (Ott et al., 2019) library. For sentiment analysis, we set the learning rate to 1e-05 and updated the model parameters with the Adam optimizer. For contrastive information valuation, the margin $\lambda$ in Eq. 7 was 1e-02. We set the learning rate to 3e-05 and adopted the Adam optimizer with the cosine learning rate scheduler (Loshchilov and Hutter, 2017). The minimum and maximum learning rates were 1e-08 and 3e-05.

During the training stage I, the input reviews were independently encoded, and the concatenated hidden states of the reviews were attended by the decoder to predict the summary. Following Press and Wolf (2017), the token embeddings were shared across the encoder and decoder for regularization. We used the learning rate of 3e-05 and

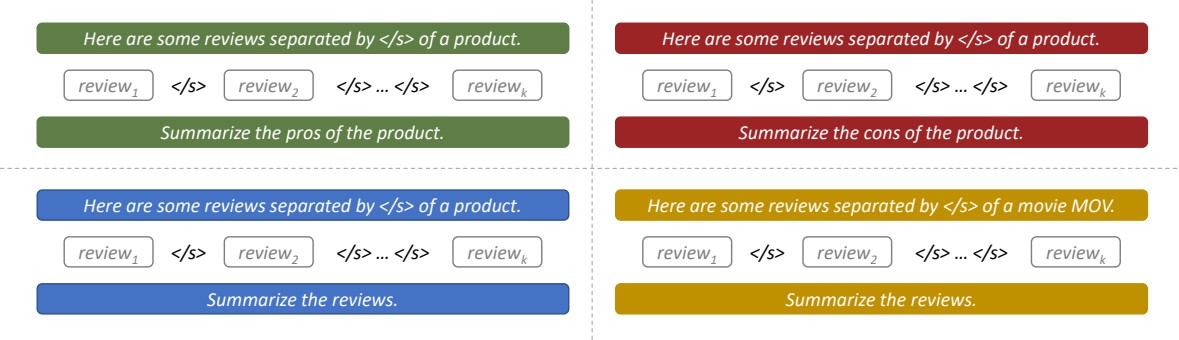

Figure 3: Prompts for LLMs in our experiments. The four prompts were used for *pros*, *cons*, and *verdict* in `AmaSum` and `Rotten Tomatoes` in order.

the Adam optimizer (Kingma and Ba, 2015) with 5,000 warmup steps for model optimization. The decoding strategy was beam search with a beam size of 5 and trigram blocking (Paulus et al., 2017).

In the training stage II, the reviews in the subset were joined with the separator  before being fed to the model, and $M = 16$ candidate summaries were collected for every training sample. We used the Adam optimizer with the same learning rate scheduler as BRIO and changed the maximum learning rate to 1e-03. We used the margin $\lambda$ of 1e-03 in Eq. 11 for all experiments. For summary generation, we used beam search of size 5. Particularly, We distinguished between the length penalty hyperparameter $\alpha$ in Eq. 12 and $lenpen$ in the beam search algorithm: the former was fixed at 2.0, and the latter differed across the targets. Other detailed hyperparameters are listed in Table 7.

For the baselines, we adapted BRIO to the task of large-scale opinion summarization. Concretely, while preparing the input, we always sampled $R_{1:K}$ out of $R_{1:N}$ and joined the $K$ reviews with the separator . The input of BRIO and CHATGPT was the optimal review subset as in Sec. 3.2 due to the maximum input length of 4096; LONGFORMER (LED-base-16384) and QG received the original review set as input. The prompts for the LLMs are presented in Fig. 3.

## D    Hyperparameter Selection

The hyperparameter $K$ has a significant impact on the performance of SUBSUMM. In this paper, we followed the setting of the baseline model SEL-SUM and inherited $K = 10$ on `AmaSum` dataset for comparability. Before working on RT dataset, we had conducted experiments with varying $K$ and random review subsets to explore the best value.

| Value | R-1 | R-2 | R-L |
|-------|-----|-----|-----|
| $K = 6$ | 21.36 | 4.30 | 15.87 |
| $K = 8$ | 22.16 | 4.89 | 16.62 |
| $K = 10$ | **23.20** | **5.56** | **17.28** |
| $K = 12$ | 22.54 | 5.40 | 16.58 |

Table 8: Results of the experiments with varying $K$ and random review subsets on RT dataset. The highest scores are shown in bold.

From the results in Table 8, it can be inferred that a too-small value of $K$ can cause information deficiency, and a too-large one may introduce the sparsity problem even after the review selection, so we didn't change the value of $K$.

## E    Human Evaluation

BWS is known to produce more reliable results than raking scales (Kiritchenko and Mohammad, 2017) and is widely used in opinion summarization studies. We randomly selected 30 samples from the *pros*, *cons*, and *verdict* partition of `AmaSum`'s test set severally and recruited 6 volunteers. The volunteers were asked to choose one best and one worst summary from four summaries for three criteria and report the confidence of their choices. For each volunteer's response, the best model received +1, the worst model received -1, and the rest of the models received 0 scores. Taking the confidence as weight, the scores of 6 volunteers were weighted and summed to get the final scores.

About the criteria, *Informativeness* tells if the summary presents opinions about specific aspects of the entity in the round, *Coherence* measures how easy the summary is to read and reflects if the summary follows a natural ordering of facts,

| Method | R-1 | R-2 | R-L |
|---|---|---|---|
| *Sampling Strategy* | | | |
| RAND | 22.73 | 5.27 | 16.80 |
| SENTI-RAND | 22.76 | 5.36 | 16.81 |
| SENTI-INFO | 23.44 | 5.84 | 17.25 |
| SENTI-RAND-INFO | **23.99** | **6.09** | **17.54** |
| *Training Scheme* | | | |
| SUBSUMM | 24.96 | 6.66 | 19.08 |
| w/o Stage I | 23.72 | 5.16 | 18.05 |
| w/o Stage II | 23.99 | 6.09 | 17.54 |
| RAND in Stage I | 25.40 | 6.74 | **19.13** |
| RAND in Stage II | **25.46** | **6.77** | 18.90 |

Table 9: Results of analysis experiments on RT. R-1/2/L are the ROUGE-1/2/L F-1 scores, and the highest scores in both blocks are shown in bold.

and *Non-Redundancy* measures the repetitions and unnecessary contents in the summary.

# F  Experiment Results

The results of the analysis experiments on RT are reported in Table 9. We list a set of example summaries from SUBSUMM and other baselines on the AmaSum dataset in Table 10.

| Method | Pros | Cons | Verdict |
|---|---|---|---|
| GOLD | A quality pumice stone crafted from 100% pure volcanic lava. Has a good shape, making it easy to hold and apply pressure to hard skin. Use the handy rope loop to hang the stone out to dry after use. | Some users find the coarseness rough and takes time to get used to. | Comfortable to hold and use, this all natural volcanic pumice stone works hard on calluses and tough skin. |
| SELSUM | Made of natural materials that are easy to clean. Comes with a rope attached to hang in the shower for easy use. | Some users find it difficult to remove dead skin while it's in use. May not remove all dead skin on the feet, but it's a good choice for those with dry skin. | A good choice if you're looking for a natural pumice stone that's easy to hold and use. |
| BRIO | A before and after, it was a bit embarrassing, and my feet were a few deep cracks for a few scrubs my feet are gone and a couple of scrubs | Users report that the pumice stone is not as smooth as they would like. Some reports of the stone breaking. | Pumice stone is the real deal and I appreciate that. using it for the first time is a great experience. |
| CHATGPT | Pumice Valley's natural pumice stone is effective in removing dead skin, comes with a handy rope for easy use, and has a fine grain that doesn't rip | The stone is not sharp or rough enough to remove dry skin, it may be chipped or in poor shape, it can | The pumice stone is effective in removing dead skin and smoothing heels, but may not work well on existing calluses. |
| SUBSUMM | Natural lava stone that's easy to use and easy to clean. Comes with a rope for hanging in the shower. Comes in a variety of sizes. Made from lava. | May be too rough for some users. Some users find it difficult to remove the calluses from the stone. May not remove all calluses. | You're looking for a pumice stone that's easy to use and clean, this is the one to buy. |

Table 10: Example summaries from SUBSUMM and other baselines on AmaSum. Contents that coincide with the reference summaries or appear erroneous for opinion summarization are highlighted.