# OpenReview forum: "Large-Scale and Multi-Perspective Opinion Summarization with Diverse Review Subsets"
_EMNLP/2023/Conference — EMNLP 2023 Findings_

### Official Review · Reviewer_AegH · 2023-08-02

**Paper Topic And Main Contributions:** 1. This paper proposes a framework ca…
**Soundness:** 3

**Excitement:**

3: Ambivalent: It has merits (e.g., it reports state-of-the-art results, the idea is nice), but there are key weaknesses (e.g., it describes incremental work), and it can significantly benefit from another round of revision. However, I won't object to accepting it if my co-reviewers champion it.

**Missing References:**

1. Suyu Ge, Jiaxin Huang, Yu Meng, Jiawei Han. FineSum: Target-Oriented, Fine-Grained Opinion Summarization. WSDM 2023.

**Questions For The Authors:**

1. Have you tested SUBSUMM on any other datasets beyond online reviews to prove its generalization ability?
2. The sampling strategies use sentiment analysis and ROUGE with reference summaries. How would the framework adapt to a purely unsupervised setting?
3. What is the inference time of SUBSUMM on large review sets? Is it applicable for real-time usage?
4. What were the main challenges faced in training the 2-stage model? Did you try any other scheduling or curriculum strategies?

**Reasons To Accept:**

1. It addresses an important problem of summarizing large volumes of opinion text from different perspectives, which is useful for many real-world applications like marketing analysis and decision making.
2. The proposed framework SUBSUMM achieves new state-of-the-art results on two benchmark datasets, and the in-depth analysis offers insights into the contribution of different components of the framework.

**Reasons To Reject:**

1. The problem setup itself is somewhat artificial - summarizing a very large set of reviews (average 76-74 reviews per product) when most real products would not have that many reviews.
2. There is no component in the framework that specifically models aspect information which is crucial for opinion summarization. It is mainly driven by ROUGE scores.
3. The two-stage training technique feels a bit ad-hoc. Ideally, the model should be able to learn from all data rather than just optimal subsets in later stages. Also, no mechanisms are introduced to reduce repetition and ensure diversity in the generated summaries.

**Reproducibility:**

3: Could reproduce the results with some difficulty. The settings of parameters are underspecified or subjectively determined; the training/evaluation data are not widely available.

**Reviewer Confidence:**

4: Quite sure. I tried to check the important points carefully. It's unlikely, though conceivable, that I missed something that should affect my ratings.

---

> ### Author Rebuttal · Authors · 2023-08-28
>
> Thank you for your valuable comments!
> ***
> ***Responses to the reasons to reject***
>
> * *Response to R1*
>
> &emsp; Thanks a lot for your attention to our research problem. Though the most commonly-used online review datasets are *Yelp* and *Amazon* with 8 reviews per product, we still observe plentiful products with a large number of reviews on some e-commerce platforms. For example, a fidget spinner (ASIN: B08HWDXFMS) on Amazon has 126 reviews, and a restaurant on Yelp could have thousands of comments. We think our method with information selection design can better facilitate reading in such scenarios.
>
> * *Response to R2*
>
> &emsp; (1) Our work is an attempt to utilize the evaluation metrics to take the place of the aspect and sentiment terms when guiding the summarizer. Specifically, we integrate ROUGE-related information into the review sampling strategies and the training loss. In Table 1 & 2, *SubSumm* establishes new SOTA on both benchmarks, outperforming the baseline *SelSum* which explicitly models the aspect information, which suggests the rationality of this trial.
>
> &emsp; (2) As mentioned in our conclusion and limitations, we agree that the model performance would be further improved if the evaluation metrics and traditional aspect information can be integrated for guidance. We plan to explore it in the future.
>
> * *Response to R3*
>
> &emsp; (1) As the number of the input reviews increases, a large part of the reviews would become dispensable due to data sparsity. Therefore, selecting valuable subsets of reviews could mitigate the sparsity problem and reduce the computation cost.
>
> &emsp; (2) We found that diversity is not frequently concerned in the field, both general summarization, for example,
>
> > Z-Code++: A Pre-trained Language Model Optimized for Abstractive Summarization (He et al., ACL 2023)
>
> > Revisiting the Gold Standard: Grounding Summarization Evaluation with Robust Human Evaluation (Liu et al., ACL 2023)
>
> and opinion summarization, for example,
>
> > Attributable and Scalable Opinion Summarization (Hosking et al., ACL 2023)
>
> > From Key Points to Key Point Hierarchy: Structured and Expressive Opinion Summarization (Cattan et al., ACL 2023)
>
> &emsp; (3) From the results, no serious issues of repetition or diversity are found. You may refer to the human evaluation results of *Informativeness* and *Non-Redundancy* in Table 3. Besides, we computed the *Distinct-N* scores of our method and part of the baselines (supervised and LLM-related) on *AmaSum*. *Distinct-N* is a metric that measures the diversity of a sentence by counting distinct n-grams in it and penalizing the sentence with lots of repeated words. It was proposed by the following paper:
> > A Diversity-Promoting Objective Function for Neural Conversation Models (Li et al., NAACL 2016)
>
> And here are the results:
>
> &emsp;|Pros|&emsp;|Cons|&emsp;|Verd|&emsp;
> -|-|-|-|-|-|-
> Method|D-1|D-2|D-1|D-2|D-1|D-2
> SelSum|**22.05**|**67.22**|23.57|68.25|**24.91**|**73.70**
> BRIO|$\underline{\text{16.70}}$|$\underline{\text{57.77}}$|20.88|$\underline{\text{65.16}}$|22.07|$\underline{\text{69.21}}$
> SubSumm|17.33|59.15|21.05|65.83|22.52|70.18
> QG|20.68|65.17|**23.87**|69.41|22.52|69.48
> ChatGPT|17.13|65.05|$\underline{\text{20.23}}$|**69.83**|$\underline{\text{20.94}}$|70.20
>
> where the highest scores are shown in bold and the lowest scores are underlined. *SubSumm* shows comparable diversity among the powerful baselines. Especially, it performs better than *BRIO* which has no special selection mechanism.
>
> &emsp; (4) When training the summarizer, we introduce stochastic factors to the optimal sampling strategy (i.e., *Sentiment-Information Ranking (modified)* in L340), where we perform weighted sampling to balance the diversity and quality of the selected review subsets.
> ***
> ***Responses to the questions***
> * *Response to Q1*
>
> &emsp; Since the current target of opinion summarization is mainly epitomizing the online reviews, we have not tried other datasets yet. We are planning to scale our study to other domains.
>
> * *Response to Q2*
>
> &emsp; The rankings of the reviews in the “information valuation” step should rely on some reference-free metric rather than ROUGE. Moreover, the training scheme is to be replaced as it is fully supervised.
>
> * *Response to Q3*
>
> &emsp; The average inference speed is less than 10 seconds per sample  (sampling + summarization) on our devices, which is similar to the baseline *SelSum* and acceptable for practical use.
>
> * *Response to Q4*
>
> &emsp; (1) During training, the biggest challenge was the dilemma between the limited model scale and the massive input reviews. We come up with a set of flexible review sampling strategies to reduce the input volume and maintain valuable information, which forms one of the contributions of our paper.
>
> &emsp; (2) We haven’t tried other training strategies yet, but we are going to experiment with more training schemes and model architectures in our future work.
>
> ***
> ***Others***
> * We are going to cite the *FineSum* paper in our revision. Thank you!

---

### Official Review · Reviewer_DG5P · 2023-08-05

**Soundness:** 3

**Excitement:**

3: Ambivalent: It has merits (e.g., it reports state-of-the-art results, the idea is nice), but there are key weaknesses (e.g., it describes incremental work), and it can significantly benefit from another round of revision. However, I won't object to accepting it if my co-reviewers champion it.

**Missing References:**

- [1] [Longformer: The Long-Document Transformer](https://arxiv.org/abs/2004.05150) (Beltagy et al., arXiv)
- [2] [Big Bird: Transformers for Longer Sequences](https://proceedings.neurips.cc/paper/2020/file/c8512d142a2d849725f31a9a7a361ab9-Paper.pdf) (Zaheer et al., NeurIPS 2020)
- [3] [DYLE: Dynamic Latent Extraction for Abstractive Long-Input Summarization](https://aclanthology.org/2022.acl-long.118) (Mao et al., ACL 2022)
- [4] [Long Document Summarization with Top-down and Bottom-up Inference](https://aclanthology.org/2023.findings-eacl.94) (Pang et al., Findings 2023)

**Paper Topic And Main Contributions:**

- This paper performs supervised opinion summarization with a long input setting. The study proposes a two-stage approach to perform the task. The experimental results show that the proposed method improves over the strong supervised opinion summarization baselines.

**Questions For The Authors:**

- I think this study is a really nice attempt. I'm really sad this is not an ARR submission. However, if you can address all the concerns raised in the reasons to reject during the rebuttal, I would consider raising the score.

# after rebuttal
Thanks for taking your time! I updated the score.

**Reasons To Accept:**

- It shows improvement compared to the strong supervised opinion summarization baselines.
- It performs human evaluation using best-worst scaling (which is preferable) and demonstrates the effectiveness of the proposed system.

**Reasons To Reject:**

- The biggest issue with this study is the lack of comparison with long-document summarization studies, such as sparse attention-based methods and reduce-then-generate approaches.
- I calculated the dataset statistics for your specific datasets and found that sparse attention-based encoder-decoder models, which often take 16k tokens as input, can handle all the reviews as input.
- So, you should try the long-document summarization methods for a more comprehensive comparison.

- I feel the BRIO and ChatGPT baselines are a bit unfair. Randomly sampled input is used for them. For training BRIO, I believe you should use an oracle set of reviews as input and perform a pipeline-based method for inference. For ChatGPT, you should use better retriever for the sampling stage.
- As of 2023, I feel it is hardly acceptable to show a ROUGE score and claim that it is a superior model to ChatGPT. It is evident that ChatGPT is not aligned with the training set distribution, making it hard to achieve high ROUGE scores. However, with human evaluation, it can perform better. So, it feels a bit suspicious to claim that the proposed method is better than ChatGPT.

**Reproducibility:**

3: Could reproduce the results with some difficulty. The settings of parameters are underspecified or subjectively determined; the training/evaluation data are not widely available.

**Reviewer Confidence:**

4: Quite sure. I tried to check the important points carefully. It's unlikely, though conceivable, that I missed something that should affect my ratings.

---

> ### Author Rebuttal · Authors · 2023-08-28
>
> Thank you for your constructive feedback!
> ***
> ***Responses to the reasons to reject***
>
> * *Response to R1-R3*
>
> &emsp; (1) Our work’s main contribution is leveraging explicit information selection mechanisms to tackle extensive input reviews. To emphasize the selective ability of the proposed framework, we prioritize *SelSum* and *BRIO*, which follow a similar select-then-summarize pipeline, for comparison.
>
> &emsp; (2) We had considered LED-base-16384 before, but we are sorry for not having sufficient computational resources then.
>
> &emsp; (3) Thanks for your attention to our experiment data! In an effort to make up for this limitation, we rented additional GPUs and fine-tuned a `LED-base-16384` model. The results are as follows.
>
> &emsp;&emsp;(i) Rotten Tomatoes
> Method|R-1|R-2|R-L
> -|-|-|-
> Longformer|24.96|6.34|18.40
> SubSumm|24.96|6.66|19.08
>
> &emsp;&emsp;(ii) AmaSum
> &emsp;|&emsp;|Pros|&emsp;|&emsp;|Cons|&emsp;|&emsp;|Verd|&emsp;
> -|-|-|-|-|-|-|-|-|-
> Method|R-1|R-2|R-L|R-1|R-2|R-L|R-1|R-2|R-L
> Longformer|22.40|4.71|15.36|14.68|2.53|11.62|22.56|4.83|17.08
> SubSumm|26.25|4.96|24.18|16.72|3.00|14.80|25.36|5.04|19.58
>
> It can be seen that as the input text gets longer (from *RT* to *AmaSum*), the edge of *SubSumm* over *Longformer* appears more remarkable. Intuitively, *Longformer*’s sparse attention mechanism plays an implicit selection role, while the sampling strategies of *SubSumm* consider the sentiment tendency and information value, which is more explicit and task-related. Therefore, our method is better than *Longformer* in handling massive reviews. We will supplement the results and the analysis in our revision.
>
> * *Response to R4*
>
> &emsp; (1) As in L968, the input of *ChatGPT* is the same as our summarizer, which is the optimal review subset mentioned in Sec. 3.2, L376.
>
> &emsp; (2) Unfortunately there are no other oracle subsets of the *AmaSum* or *RT* released. We trained *BRIO* on the optimal review subsets in our paper, and the results are as follows, which are better but still surpassed by our method. We are going to report the results in our revision.
>
> &emsp;&emsp;(i) Rotten Tomatoes
> Method|R-1|R-2|R-L
> -|-|-|-
> BRIO (oracle)|23.72|5.16|18.05
> SubSumm|24.96|6.66|19.08
>
> &emsp;&emsp;(ii) AmaSum
> &emsp;|&emsp;|Pros|&emsp;|&emsp;|Cons|&emsp;|&emsp;|Verd|&emsp;
> -|-|-|-|-|-|-|-|-|-
> Method|R-1|R-2|R-L|R-1|R-2|R-L|R-1|R-2|R-L
> BRIO (oracle)|25.48|4.58|23.50|16.65|2.94|14.60|24.93|4.78|19.44
> SubSumm|26.25|4.96|24.18|16.72|3.00|14.80|25.36|5.04|19.58
>
> * *Response to R5*
>
> &emsp; (1) Although ROUGE has a long history and apparent limitations, it is very frequently adopted for evaluation of the opinion summarization methods.
>
> &emsp; (2) Before the human evaluation, we compared the summaries generated by different systems preliminarily. The summaries of *ChatGPT* are fluent, yet they do suffer from more severe hallucination than our model, which may compromised their ROUGE scores.
>
> &emsp; (3) We have conducted an extra human evaluation to compare *SubSumm* with *ChatGPT*. 50 cases were randomly sampled from the two test sets (15 cases from each partition in *AmaSum* and 5 cases from *RT*). The recruited 6 volunteers were asked to conduct pair-wise comparisons without the reference summaries. The results are as follows:
>
> Data|SubSumm vs. ChatGPT
> -|-
> AmaSum - cons|66.67% vs. 33.33%
> AmaSum - pros|65.56% vs. 34.44%
> AmaSum - verd|88.89% vs. 11.11%
> Rotten Tomatoes|70.00% vs. 30.00%
>
> It seems that the raters prefer the summaries generated by *SubSumm*. An obvious issue of *ChatGPT* is that it can’t control the output length within a few calls when the input is overlong. Consequently, most of the summaries generated are either excessively long or abruptly truncated with the maximum length argument fixed. We will report the results and the corresponding analysis in our revision. For (2) and (3) you may refer to Table 8 to see some supporting cases.
>
> ***
> ***Others***
> * We will supplement the references with all the mentioned papers. Thanks!
> * Again, we are grateful to receive your affirmation, and we would appreciate it if you can reconsider your score.

---

### Official Review · Reviewer_RBcE · 2023-08-06

**Soundness:** 3

**Excitement:**

3: Ambivalent: It has merits (e.g., it reports state-of-the-art results, the idea is nice), but there are key weaknesses (e.g., it describes incremental work), and it can significantly benefit from another round of revision. However, I won't object to accepting it if my co-reviewers champion it.

**Missing References:**

Line 130 - “These methods work well…” - Is this an opinion of the paper. Can you add a previous work that shows that the previous works dont work when the number of reviews increase

**Paper Topic And Main Contributions:**

In the realm of opinion summarization, one of the primary challenges lies in creating comprehensive summaries that encompass diverse perspectives. The task becomes even more complex when dealing with a large number of reviews. However, this paper introduces a novel approach to address this issue by leveraging sentiment analysis and contrasting information from various reviews to identify the most suitable ones for each type of summary. The proposed model undergoes two stages of training. In the first stage, it is trained using Maximum Likelihood Estimation (MLE) to generate the initial summaries. Subsequently, in the second stage, the model is further refined by utilizing its capability to produce reasonable summaries, and it is trained on its own generated outputs. This innovative method shows promise in enhancing opinion summarization and streamlining the selection of reviews for different perspectives, ultimately leading to more effective and insightful summaries.

**Questions For The Authors:**

- Clarification on K Value in Experiments:  The paper would benefit from explicitly stating the value of K used in the experiments within the main body of the text. If this information is currently relegated to the appendix, it should be brought forward to the main part of the paper for better accessibility and understanding.
- The paper lacks clarity on how different subsets of reviews are chosen from Rotten Tomatoes data. It's also unclear which category the summary belongs to—pros, cons, or verdict. These aspects require further explanation for better understanding.
- Experiments involving retrieve and generate models, while not strictly mandatory, have become increasingly pertinent in the current age. The reviewer is understandably curious about your paper's positioning within the framework of "Retrieve relevant context and generate." To address this curiosity, it would be valuable if you could furnish experiments demonstrating the application of your retrieval strategy on a Language Model (LM) like LLaMa and observe the resulting outcomes. Such experiments would serve to validate the relevance and effectiveness of your retrieval strategies independently of the specific language model employed

**Reasons To Accept:**

- ***Refreshing idea***
This idea presents a refreshing and novel approach by combining various techniques to achieve opinion summarization.
- ***Good empirical analysis***

    The paper conducts a robust empirical analysis, considering a substantial number of baselines and recent advances for comparison. Furthermore, the authors conduct thorough ablation studies, effectively justifying their choice for the final system design.

**Reasons To Reject:**

- Mismatch between motivation and experiments - One notable aspect where the paper falls short is the absence of comparisons with models capable of handling long context, such as the Longformer or any other recently released model with extended context length. Including such comparisons would have enriched the analysis and strengthened the overall contribution of the study.
- Experiemnts with different values of K and N - The paper lacks experiments for different values of K, which is a significant limitation. By exploring various values of K, the authors could have gained valuable insights into the performance and behavior of their method under different settings. Additionally, it is essential to assess how well the proposed method performs when N (the number of samples or data points) is higher compared to other existing methods, as this was a primary motivation behind the research. By conducting experiments with varying N, the authors could have demonstrated the scalability and effectiveness of their approach and provided a more comprehensive evaluation of its performance in real-world scenarios. This absence of experimentation limits the paper's ability to fully support its main claims and conclusions. Including experiments with different values of K and diverse N settings would significantly enhance the paper's scientific rigor and strengthen its contribution to the field.

**Reproducibility:**

4: Could mostly reproduce the results, but there may be some variation because of sample variance or minor variations in their interpretation of the protocol or method.

**Reviewer Confidence:**

4: Quite sure. I tried to check the important points carefully. It's unlikely, though conceivable, that I missed something that should affect my ratings.

**Typos Grammar Style And Presentation Improvements:**

The first line in the introduction is too cumbersome to read. I think you can remove the heavy sounding words and simplify it.

---

> ### Author Rebuttal · Authors · 2023-08-28
>
> Thank you for your thoughtful comments!
> ***
> ***Responses to the reasons to reject***
>
> * *Response to R1*
>
> &emsp; (1) Our work’s main contribution is leveraging explicit information selection mechanisms to tackle extensive input reviews. To emphasize the selective ability of the proposed framework, we prioritize *SelSum* and *BRIO*, which follow a similar select-then-summarize pipeline, for comparison.
>
> &emsp; (2) We had considered LED-base-16384 before, but we are sorry for not having sufficient computational resources then.
>
> &emsp; (3) In an effort to make up for this limitation, we rented additional GPUs and fine-tuned a `LED-base-16384` model. The results are as follows.
>
> &emsp;&emsp;(i) Rotten Tomatoes
> Method|R-1|R-2|R-L
> -|-|-|-
> Longformer|24.96|6.34|18.40
> SubSumm|24.96|6.66|19.08
>
> &emsp;&emsp;(ii) AmaSum
> &emsp;|&emsp;|Pros|&emsp;|&emsp;|Cons|&emsp;|&emsp;|Verd|&emsp;
> -|-|-|-|-|-|-|-|-|-
> Method|R-1|R-2|R-L|R-1|R-2|R-L|R-1|R-2|R-L
> Longformer|22.40|4.71|15.36|14.68|2.53|11.62|22.56|4.83|17.08
> SubSumm|26.25|4.96|24.18|16.72|3.00|14.80|25.36|5.04|19.58
>
> It can be seen that as the input text gets longer (from *RT* to *AmaSum*), the edge of *SubSumm* over *Longformer* appears more remarkable. Intuitively, *Longformer*’s sparse attention mechanism plays an implicit selection role, while the sampling strategies of *SubSumm* consider the sentiment tendency and information value, which is more explicit and task-related. Therefore, our method is better than *Longformer* in handling massive reviews. We will supplement the results and the analysis in our revision.
>
> * *Response to R2*
>
> &emsp; (1) Considering the comparability, we follow the setting of the baseline model *SelSum* and inherit K=10 on the *AmaSum* dataset.
>
> &emsp; (2) Before working on the *RT* dataset, we had conducted a group of experiments with varying K and random review subsets to explore the best value:
> Value|R-1|R-2|R-L
> -|-|-|-
> K=6|21.36|4.30|15.87
> K=8|22.16|4.89|16.62
> K=10|23.20|5.56|17.28
> K=12|22.54|5.40|16.58
>
> It can be inferred that a too small value of K can cause information deficiency, and a too large one may introduce the sparsity problem even after the selection, so we didn’t change the value of K. We will add the results to our revised paper.
>
> &emsp; (3) While designing the review sampling strategies, both criteria are calculated in a point-wise manner. The loss terms used for fine-tuning the LMs are either point-wise or pair-wise. Hence our method has better scalability than other list-wise approaches theoretically.
>
> &emsp; (4) To our knowledge, we have selected the two datasets with largest N among all the opinion summarization datasets. After preprocessing, the value of N varies with the samples from 10 to 100 in both datasets, which has already covered most situations. The SOTA performance can reflect acceptable scalability of our approach.
> ***
> ***Responses to the questions***
> * *Response to Q1*
>
> &emsp; Thanks a lot for your suggestion. We will move the clarification of the value of K (K=10) from the appendix (L937) to the implementation details in Sec. 4.1.
>
> * *Response to Q2*
>
> &emsp; Thank you! We have stated that the *verdict* summary is similar to general opinion summary in the dataset introduction (L390), and we will modify it to clarify that the data in *RT* and the *verdict* partition of *AmaSum* was equally treated in our experiments.
>
> * *Response to Q3*
>
> &emsp; (1) So far as we know, the most cutting-edge retrieve-and-generate application is the combination of advanced search engine and Large Language Model (LLM). Limited by the research conditions, we are not able to experiment with such foundation models. However, our proposal provides an idea of orientally retrieving information according to multiple criteria, then continuously improving model performance by manipulating different information, which we think to be applicable and promising in this framework.
>
> &emsp; (2) We have justified the effectiveness of the review sampling strategy itself in our experiments.
>
> &emsp; &emsp; (i) In the comparison experiments (Table 1 & 2), *SubSumm*, *SelSum*, and *BRIO* employ the same LM (i.e., *BART*), yet our method outperforms the other two, evidencing that the performance gains from the framework are model-agnostic.
>
> &emsp; &emsp; (ii) In the ablation study (Table 4 & 7), we first demonstrate that the proposed sampling strategies differ in quality (top half of the tables), then verify the contribution of the optimal strategy in both training stages (the last two rows of the tables).
>
> &emsp; &emsp; (iii) Moreover, the supplementary experiments of *Longformer* also show that equipped with our sampling strategy set and the corresponding training scheme, the smaller LM can defeat the larger LM.
> ***
> ***Others***
> * We will cite the recent papers studying opinion summarization with a large number of reviews.
> * We are going to simplify the first sentence of the introduction in our revision. Thanks for reminding us!

---

### Meta-Review · Area_Chair_72Md · 2023-09-14

**Recommendation:** 3

**Metareview:**

The paper introduces a new supervised opinion summarization model for a large number of reviews. A new sampling strategy based on sentiment analysis and contrastive information evaluation is introduced for the selection of high-quality review subsets from a large input set.  The study suggests a two-stage process for carrying out the task: training using MLE to generate the initial summaries and then refining the model with contrastive learning to assign a higher probability to better candidate summaries. The results of the experiments demonstrate that the suggested method outperforms the robust supervised opinion summarization baselines.

Pros:
1. New sampling strategy for selecting the best candidates for review summarization
2. A novel two-stage training scheme
3. The experimental results, including human evaluation using best-worst scaling, show the superiority of the proposed system compared to the strong supervised opinion summarization baselines.

Cons:
Some reviewers noticed missing comparisons with LLMs designed for long inputs. The authors ran experiments and reported the results in their rebuttals.
The authors responded to every reviewer with detailed letters. Two reviewers increased their scores.

---

### Decision · Program_Chairs · 2023-10-07

**Decision:**

Accept-Findings

**Comment:**

The paper introduces a new supervised opinion summarization model for a large number of reviews. A new sampling strategy based on sentiment analysis and contrastive information evaluation is introduced for the selection of high-quality review subsets from a large input set.  The study suggests a two-stage process for carrying out the task: training using MLE to generate the initial summaries and then refining the model with contrastive learning to assign a higher probability to better candidate summaries. The results of the experiments demonstrate that the suggested method outperforms the robust supervised opinion summarization baselines.

Pros:
1. New sampling strategy for selecting the best candidates for review summarization
2. A novel two-stage training scheme
3. The experimental results, including human evaluation using best-worst scaling, show the superiority of the proposed system compared to the strong supervised opinion summarization baselines.

Cons:
Some reviewers noticed missing comparisons with LLMs designed for long inputs. The authors ran experiments and reported the results in their rebuttals.
The authors responded to every reviewer with detailed letters. Two reviewers increased their scores.